# Examining SEB skills' incremental validity over personality traits in predicting academic achievement

Hee Jun Yoon[1]*, Brent W. Roberts[1], Madison N. Sewell[2], Christopher M. Napolitano[2], Christopher J. Soto[3], Dana Murano[4], Alex Casillas[4]

1 Department of Psychology, University of Illinois, Champaign, Illinois, United States of America,
2 Department of Educational Psychology, University of Illinois, Champaign, Illinois, United States of America,
3 Department of Psychology, Colby College, Waterville, Maine, United States of America, 4 ACT, Inc., Iowa City, Iowa, United States of America

* yoon14@illinois.edu

**Data Availability Statement:** https://osf.io/uewav/?view_only=71c5aed524cc477194a76d22e69b7a6e.

**Funding:** The authors received no specific funding for this work.

## Abstract

Personality traits and social, emotional, and behavioral (SEB) skills share the same behavioral referents, but whereas traits refer to a person's typical or average performance, skills refer to their capacity or maximal performance. Given their shared behavioral foundations, an important question to address is whether personality traits and SEB skills independently predict important outcomes. In this study (N = 642), we examined whether subscales of the Behavioral, Emotional, and Social Skills Inventory (BESSI), a measure of SEB skills, provided incremental validity in the prediction of the ACT composite score, an important academic outcome for American adolescents, over the Big Five personality traits. Consistent with our expectations, on average, SEB skills showed stronger associations with ACT achievement scores than personality traits. Moreover, SEB skills added incremental validity over and above personality traits in predicting ACT achievement scores. The findings reinforce the importance of conceptually distinguishing and measuring traits and skills.

## Introduction

Research has shown that personality traits are associated with important life outcomes, such as job performance [1], longevity [2], and subjective well-being [3]. In the academic domain, openness to experience is the strongest personality predictor of standardized test scores for high school students [4], and conscientiousness and openness are key predictors of academic performance across different academic levels from primary to tertiary education [5].

Recently, there has been renewed attention to the potential distinction between personality traits and skills, with the goal of better differentiating these two ways of conceptualizing individual differences and understanding their implications for success in life [6]. Although a recent focus, this distinction is not new. There is a longstanding position in personality psychology [7] that focused on the conceptual distinction between behavioral traits (e.g., the interpersonal circumplex, the Big Five) and capabilities [8]. The consensus definition of personality

**Competing interests:** The authors have declared that no competing interests exist.

traits is the relatively enduring, automatic patterns of thoughts, feelings, and behaviors that arise in isomorphic situations. It is widely accepted that, at a minimum, traits can be organized into the Big Five: Extraversion, agreeableness, conscientiousness, emotional stability, and openness to experience. More concretely, personality traits are what one typically does when faced with the same circumstances. For example, an extraverted person will more regularly initiate conversations, talk with strangers, and speak up in a crowded room. The power of the idea of personality traits is the regularity of the concept. It is the persistence of personality traits which putatively gives them their predictive power as they have a continuous effect on a person's social and occupational conditions.

Various definitions of "social, emotional, and behavioral skills" (SEB) exist, though there is not a clear consensus on how to define this term. The definitions of social and emotional skills appear to split into two broad categories–those that attempt to include the wide range of constructs under an umbrella term and those that reflect the colloquial meaning of skill. As for the first definition, social and emotional skills are "individual characteristics that (a) originate in the reciprocal interaction between biological predispositions and environmental factors; (b) are manifested in consistent patterns of thoughts, feelings, and behaviors; (c) continue to develop through formal and informal learning experiences; and (d) influence important socio-economic outcomes throughout the individual's life" [9 p. 279]. Another example definition is "the broad range of malleable skills that enable individuals to navigate interpersonal and social situations effectively" [10 p. 5]. According to Duckworth and Yeager [11 p. 239], SEB skills have five things in common: they are "(a) conceptually independent from cognitive ability, (b) generally accepted as beneficial to the student and to others in society, (c) relatively rank-order stable over time in the absence of exogenous forces (e.g., intentional intervention, life events, changes in social roles), (d) potentially responsive to intervention, and (e) dependent on situational factors for their expression". These definitions of SEB skills are conspicuous for one reason in this context–they are all inclusive of, if not consistent with the definition of personality traits. For example, recent research has shown that despite their "relatively enduring" nature, personality traits are both changing and changeable [12, 13].

If malleability is not the key distinction between traits and skills, what is? Conceptually, the most important difference between traits and skills is the distinction between proclivities and capabilities. Personality traits reflect the propensity or proclivity to enact thoughts, feelings, or behaviors. They reflect the discernable mode or middle of the density distribution of intrinsically variable behaviors [14]. On average an extraverted person will be more talkative than not, but there will also be situations where they say little and other situations in which they dominate a discussion even more than they typically do. The mode or middle of the distribution is assumed to be the most likely manifestation of extraversion and the best representation of what that person will be likely to do over time and across situations.

Skills on the other hand, represent what a person is capable of doing. In terms of power tests and physical abilities, this distinction is clear. In tests of cognitive ability or achievement, the task is to perform at one's limits of ability–to try one's hardest to show what one is maximally capable of doing. Similarly, tests of physical ability represent the best an athlete can do on any given day. The analogy holds for SEB skills also. Revisiting the extraversion example, the equivalent skill variant would be a person's capacity to engage socially with other people. While it may be that their proclivity is to be introverted, a person may still possess the capacity to "turn it on" in the moment and engage successfully with a room full of strangers. It may not be their favorite activity or one that they feel comfortable enacting, but they have the capacity to do so if called upon. In this way, the skill aspect of extraversion would be better reflected in the range of behaviors around the modal aspect of the distribution–especially the upper limit of that range. However, just the upper limit of a distribution of naturally occurring behavior

may not be exactly the same as a person's maximal capacity to perform that behavior as the demand to perform a behavior at that level may not be encountered often if at all. In this way, it would be best to approach the assessment of an SEB skill as the maximum level of the skill that the person is capable of performing when called upon to do so.

Moreover, conceptual differences between personality traits and SEB skills can manifest differently in predicting educational outcomes. Personality traits may best predict academic outcomes that reflect overall performance over time across assignments and exams, such as cumulative school grades, because personality traits represent consistent patterns of thought, feelings, and behaviors of individuals. On the other hand, SEB skills may play an especially important role in predicting standardized test scores because they represent capacities that people can purposefully employ when a specific, high-stakes situation calls for it. Therefore, even though traits and skills may both play an important role in predicting academic outcomes, they may play different roles for different outcomes.

Another feature of work on SEB skills is the lack of an overarching, organizational taxonomy. Recently, Soto and colleagues [15] argued that if traits and skills are best distinguished by the way they are assessed, not necessarily their content, that researchers could take advantage of the Big Five personality taxonomy to map out a comprehensive model of social, emotional, and behavioral (SEB) skills. Accordingly, they introduced the Behavioral, Emotion, Social Skills Inventory (BESSI)—a measure of SEB skills that captures 32 SEB skill facets nested within five broad SEB skill domains (social engagement skills, cooperation skills, self-management skills, emotional resilience skills, and innovation skills). The validation work on the BESSI demonstrated that the 32 different SEB skill facets could be reliably and validly assessed within a higher-order, five-factor structure [8]. The BESSI also showed reasonable convergence with existing measures of socioemotional strengths and competencies, such as the Tripartite Taxonomy of Character, but also rather high correlations with Big Five trait measures (average correlation of .75 between corresponding Big Five traits and BESSI skills). Although the BESSI provided incremental validity over the Big Five in the prediction of self-reported measures of strengths and competencies, there was not an opportunity to test the incremental validity of the BESSI over traditional Big Five measures for predicting more objective outcomes, such as achievement test scores.

Another study examined relationships between SEB skills, personality traits, and academic outcomes in a sample of 975 adolescents [16]. They assessed the skill (i.e., from "Not at all well" to "Extremely well") and the trait (i.e., from "Not at all often" to "Extremely often") versions using the same social, emotional, and behavioral referents using the same items. The study found that both SEB skills and traits robustly predicted academic outcomes. As predictors, skills and traits were found to be interchangeable, adding no incremental predictive value over each other in predicting some academic outcomes, such as school grades and educational aspirations. However, for high-stakes scenarios like standardized test performance, skills had a modest but statistically significant incremental validity over traits. The study's findings support the utility of assessing SEB skills for predicting various types of outcomes, particularly those that occur in high-stakes situations. One limitation of the study is the potential for common method bias, which may have inflated the observed relationship between skills and traits due to the use of identical assessment items. The present study aims to address this gap by measuring the Big Five personality traits in a traditional manner.

Therefore, in order to provide a test of the incremental validity of an SEB skills conceptualization of Big Five content, we conducted a study in which we used a selection of SEB skills to predict objective academic performance above and beyond the Big Five personality traits. Specifically, we used a sample of high school participants who took the ACT achievement test and also voluntarily completed measures of personality traits and several BESSI facets chosen to

parallel the Big Five traits in terms of their social, emotional, and behavioral referents. We used these data to determine if the subset of BESSI scales provide incremental validity over the Big Five traits when predicting overall ACT achievement test scores.

## Method

### Participants and procedure

2,000 US high school students who took the ACT on the February 2020 national test date received a survey invitation [17]. The first 300 respondents who completed the follow-up research to investigate non-cognitive predictors of achievement received a $10 Amazon gift card. In total, 642 US high school students completed the survey. The participants did not know the specific purpose of the study. Participation was anonymous and voluntary. The mean age was 17.3 years (SD = .63), and approximately 72.9% of the respondents were female. The sample was diverse in terms of ethnicity (63.6% White, 10.6% Hispanic/Latino, 7.8% Black/African American, 7.6% Asian, 4.5% two or more races, 3.6% prefer not to respond, 0.3% Native Hawaiian/other Pacific islander, and 0.2% American Indian/Alaska native). The R-script, codebook, and data used for the analysis can be found here: https://osf.io/uewav/?view_only=71c5aed524cc477194a76d22e69b7a6e.

### Measures

**SEB skills.**　A total of 30 items were used to assess one representative facet from each of the five SEB skill domains assessed by the BESSI: the Goal-regulation facet representing the Self-Management domain (corresponding to Big Five Conscientiousness), the Leadership Skill facet representing the Social Engagement domain (corresponding to Extraversion), the Teamwork Skill facet representing the Cooperation domain (corresponding to Agreeableness), the Abstract Thinking Skill facet representing the Innovation domain (corresponding to Openness to Experience), and the Stress Regulation facet representing the Emotional Resilience domain (corresponding to Emotional Stability). Specific facets were chosen based on their highest loadings with the corresponding SEB skill domains. Each skill was assessed using six items. Respondents were asked to rate, on a five-point scale, ranging from 1 = Not at all well (Beginner) to 5 = Extremely well (Expert), reflecting their perception of how well they could perform that behavior, reflecting their current level of expertise (e.g., *set clear goals; lead a group of people; work as part of a group; understand abstract ideas; cope with stress*). In this high school student sample, alpha reliabilities for the BESSI scales were .90 for Self-Management, .91 for Social Engagement, .88 for Cooperation, .88 for Innovation, and .93 for Emotional Resilience.

**Big Five Inventory (BFI).**　The BFI is a 44-item questionnaire designed to measure the Big Five personality domains: Extraversion, Agreeableness, Conscientiousness, Emotional Stability, and Openness to Experience [18]. Respondents rated characteristics that may or may not apply to them, and each item was on a 6-point scale ranging from 1 = Strongly disagree to 6 = Strongly agree. In the present sample, alpha reliabilities were .82 for Conscientiousness, .86 for Extraversion, .81 for Agreeableness, .78 for Openness, and .81 for Emotional Stability.

**Academic achievement.**　Academic achievement was assessed using the overall ACT composite score, which is computed from scores on the ACT English, Math, Reading, and Science sections. ACT composite scores were measured on a 0–36 scale. The mean and standard deviation for ACT composite scores for the participants were 25.94 and 5.83, respectively. The mean for ACT composite scores for the sample was above the national average of 20.6 (ACT, 2020). According to the most recent ACT technical manual, the reliability for the ACT composite score is .97, and it is predictive of college GPA and adds incremental validity above and beyond high school GPA in predicting college GPA (ACT, 2019).

**Demographic information.** We used gender, ethnicity, and family income as control variables as they are all related personality traits, SEB skills, and achievement scores. Gender and ethnicity were dummy-coded for the analysis. Family income was measured by one item asking the total combined income of parents before taxes last year, and participants chose from 1 (Less than $24,000) to 9 (More than $150,000).

## Statistical analyses

To examine the relationships between the BESSI, BFI, and ACT, we first computed descriptive statistics and correlations among the variables. To examine the incremental validity of SEB skills above and beyond the Big Five personality traits in predicting the ACT composite scores, we used multiple regression analyses. All analyses were conducted in the R programming language [19]. We used the alpha level of .05 for statistical significance. Table 1 provides precise p-values for beta coefficients to help future researchers compute effect sizes for their secondary analysis. We did not preregister our hypothesis nor methods as our study was an exploratory study examining relationships between personality traits, socio-emotional skills, and academic achievement.

## Results

### Correlations between the BESSI, BFI, and ACT

Table 2 shows the means, standard deviations, and reliabilities as well as intercorrelations among all variables. Several notable results emerged. First, there was a general pattern of medium to strong intercorrelations among measures drawn from the same instrument. There were general positive correlations among the SEB skills, the Big Five traits, and the ACT composite score. Second, the convergent correlations between the SEB skills and their Big Five counterparts ranged from .48 to .70 and averaged .59, whereas the discriminant correlations averaged .27. This pattern of substantial–but far from perfect–associations supports SEB skill's discriminant validity. This pattern supports SEB skills' discriminant validity. Third, four of the five SEB skills showed positive associations with the ACT composite score, the strongest of which was Innovation Skill ($r = .33$, $p < .05$). Similarly, but to a lesser extent, several of the Big Five traits were also positively associated with the ACT composite score, with Conscientiousness ($r = .17$, $p < .05$) and Openness ($r = .14$, $p < .05$) showing the strongest correlations. Overall, the BESSI skills (Mean $r = .20$) showed larger correlations with the ACT composite score than the Big Five trait domains (Mean $r = .10$).

### Incremental validity of SEB skills and Big Five traits

Our second set of analyses used multiple linear regression to test whether the BESSI skills provided incremental validity in the prediction of the ACT composite score beyond their Big Five counterparts while controlling for demographic variables, such as gender, race/ethnicity, and family income levels. To further control for overlapping information within each set of constructs (skills and traits), we examined the BESSI and BFI scales as sets rather than individually. Specifically, we fit a series of regression models in which academic achievement was predicted by demographic characteristics alone (Model 1); by demographic characteristics plus either the Big Five traits (Model 2a) or five BESSI skills (Model 2b); or by all predictors simultaneously (Model 3). The results of these models are presented in Table 1.

When the demographic predictors were entered alone, the regression model accounted for 14% of variance ($p < .05$) in achievement. When the BFI scales were stepped in simultaneously in model 2a, the BFI Openness ($\beta = .15$, $p < .05$) and Conscientiousness ($\beta = .16$, $p < .05$) traits positively predicted achievement, while BFI Agreeableness ($\beta = -.14$, $p < .05$) negatively

**Table 1. Testing the Incremental Validity of the BESSI and BFI scales in aggregate.**

| DV | ACT Composite | | |
|---|---|---|---|
| | Beta | SE | P-value |
| **Step1 (Model 1)** | | | |
| Gender | -.11 | .04 | .00869 |
| White/Non-White | .01 | .04 | .80763 |
| Family income | .34 | .04 | 1.05e-13 |
| R | .14 | | |
| **Step2 –BFI (Model 2a)** | | | |
| Gender | -.11 | .04 | .01561 |
| White/Non-White | .01 | .04 | .89468 |
| Family income | .32 | .04 | 3.61e-12 |
| O | .15 | .04 | .00086 |
| C | .16 | .05 | .00302 |
| E | -.02 | .05 | .73244 |
| A | -.14 | .05 | .00481 |
| ES | .04 | .05 | .43566 |
| R | .19 | | |
| Delta R over controls | .05 | | |
| **Step2—BESSI (Model 2b)** | | | |
| Gender | -.15 | .05 | .00424 |
| White/Non-White | .01 | .05 | .76226 |
| Family income | .25 | .04 | 9.42e-07 |
| Innovation | .30 | .06 | 5.31e-07 |
| Self-Management | .07 | .06 | .23497 |
| Social Engagement | -.03 | .06 | .60948 |
| Cooperation | -.03 | .06 | .53834 |
| Emotional Resilience | -.07 | .05 | .21206 |
| R | .20 | | |
| Delta R over controls | .06 | | |
| Step 3 **(Model 3)** | | | |
| Gender | -.11 | .05 | .04260 |
| White/Non-White | .02 | .05 | .71079 |
| Family income | .24 | .05 | 1.7e-06 |
| O | .09 | .06 | .10226 |
| C | .12 | .07 | .09054 |
| E | -.07 | .06 | .22825 |
| A | -.20 | .06 | .00194 |
| ES | .14 | .08 | .07776 |
| Innovation | .23 | .07 | .00075 |
| Self-Management | .04 | .07 | .52576 |
| Social Engagement | -.04 | .07 | .50618 |
| Cooperation | .06 | .07 | .40670 |
| Emotional Resilience | -.16 | .07 | .02877 |
| R | .24 | | |
| Delta R over controls and BESSI | .04 | | |
| Delta R over controls and BFI | .06 | | |

Note. Gender was dummy-coded (female = 1, male = 0) and ethnicity was dummy-coded (White = 1, non-white = 0).

**Table 2. Descriptive statistics and intercorrelations among study variables.**

| Variables | M | SD | N | 1 | 2 | 3 | 4 | 5 | 6 | 7 | 8 | 9 | 10 | 11 | 12 | 13 | 14 |
|---|---|---|---|---|---|---|---|---|---|---|---|---|---|---|---|---|---|
| 1. BFI Openness | 4.37 | .69 | 603 | .78 | | | | | | | | | | | | | |
| 2. BFI Conscientiousness | 4.42 | .72 | 603 | .24 | .82 | | | | | | | | | | | | |
| 3. BFI Extraversion | 3.75 | .94 | 603 | .21 | .16 | .86 | | | | | | | | | | | |
| 4. BFI Agreeableness | 4.57 | .71 | 603 | .22 | .42 | .22 | .81 | | | | | | | | | | |
| 5. BFI Emotional Stability | 3.51 | .86 | 603 | .18 | .36 | .27 | .34 | .81 | | | | | | | | | |
| 6. BESSI Innovation | 3.80 | .82 | 480 | .60 | .34 | .17 | .14 | .20 | .88 | | | | | | | | |
| 7. BESSI Self Management | 3.91 | .81 | 482 | .30 | .63 | .26 | .30 | .26 | .49 | .90 | | | | | | | |
| 8. BESSI Social Engagement | 3.36 | .95 | 482 | .36 | .41 | .48 | .15 | .27 | .49 | .51 | .91 | | | | | | |
| 9. BESSI Cooperation | 3.88 | .68 | 482 | .31 | .41 | .26 | .52 | .26 | .43 | .53 | .51 | .88 | | | | | |
| 10. BESSI Emotional Resilience | 3.04 | .91 | 480 | .26 | .30 | .16 | .21 | .70 | .38 | .30 | .35 | .36 | .93 | | | | |
| 11. ACT Composite | 25.94 | 5.83 | 630 | .14 | .17 | -.01 | -.05 | .11 | .33 | .24 | .18 | .16 | .09 | | | | |
| 12. Gender | | | | .03 | .07 | .07 | .12 | -.27 | -.07 | -.03 | -.06 | .02 | -.28 | -.18 | | | |
| 13. Ethnicity | | | | -.02 | .09 | -.01 | .03 | -.01 | .01 | -.02 | -.05 | -.01 | .03 | .08 | .03 | | |
| 14. Family Income | 6.01 | 2.56 | 439 | -.07 | .07 | .03 | -.02 | .05 | .10 | .09 | .10 | .11 | .03 | .36 | -.06 | .23 | |

*Note*. Reliabilities for SEB skills and personality traits are shown in the diagonal; Gender was dummy-coded (female = 1, male = 0) and ethnicity was dummy-coded (White = 1, non-white = 0).

predicted achievement. These results were aligned with results from previous studies on relationships between the Big Five traits and academic achievement [5]. The Big Five traits accounted for significant incremental validity above and beyond demographic predictors (delta R square = .05, p < .05).

When the BESSI skills were stepped in simultaneously in model 2b, only Abstract Thinking Skill ($\beta$ = .30, p < .05) positively and statistically significantly predicted the ACT Composite scale score. The BESSI skills accounted for significant incremental validity beyond demographic predictors (delta R square = .06, p < .05). Finally, when both the five BFI scales and the five BESSI scales were used to predict the ACT Composite score in model 3, the BESSI Abstract Thinking skill remained a positive predictor and the BESSI Stress Regulation skill and BFI Agreeableness trait emerged as negative predictors of achievement. Our interpretation of these latter two findings is that they are suppression artifacts, given that the sign of the relation either flipped (BESSI Stress Regulation) or the magnitude of the negative effect was enhanced (BFI Agreeableness). In other words, these results indicate that Stress Regulation skill and Agreeableness trait are less positively associated with the ACT performance than would be expected from these constructs' positive relations with Innovation skill. Moreover, the BESSI skills accounted for significant incremental validity above and beyond the model with demographic predictors and the Big Five traits only (delta R-square = .06, p < .05). Conversely, the Big Five traits provided a smaller but still significant increment beyond the model with demographic predictors and the BESSI skills (delta R-square = .04, p < .05).

Overall, the most consistent finding was that Abstract Thinking Skill, as assessed by the BESSI, was a positive, statistically significant, and incremental predictor of achievement beyond demographic characteristics and the Big Five traits. Moreover, the BESSI skills collectively added significant incremental validity above and beyond the Big Five traits in predicting academic achievement.

## Discussion

In the present research, we tested whether particular SEB skills predicted high school students' ACT scores beyond their Big Five personality trait counterparts. In this case, we pitted an

equal number of skills and trait scales selected for their conceptual overlap in the prediction of objective academic achievement. This constitutes a challenging form of incremental validity, as the BESSI and BFI scales share very similar behavioral referents and differ primarily in how they are rated. The BESSI focuses on a person's capacity to perform personality-relevant thoughts, feelings, and behaviors, while the BFI assesses people's typical patterns of thoughts, feelings, and behaviors. We suspected that SEB skills might provide incremental validity for predicting performance in a high-stakes situation, although this expectation was tempered by the high degree of overlap between the BESSI and BFI scales in prior research [6].

Our results support the conclusion that skill and trait conceptualization of personality differ enough to provide incremental information in the prediction of ACT scores, a critical high-stakes test for American adolescents. This shows that while BESSI skills are related to personality traits, they are not identical, and account for different components of the variance in high-stakes academic achievement. This finding adds to a recent and growing body of evidence that skills can provide incremental validity over traits in predicting a broad range of consequential outcomes including academic engagement and achievement, occupational interests, social behaviors and relationships, and physical and psychological well-being [6, 20–21].

Given the utility of a skills-based conceptualization of personality for educational and industry settings, practitioners might find the BESSI better suited for examining the effects of SEB skill interventions designed to improve academic performance. For example, individuals do not have to work hard or be social in all situations. Instead, they can exhibit self-regulation skills and leadership skills when situations call for it. Similarly, having teachers focus on skill development of students would be better calibrated to what is typically done in a classroom. Most curricula are structured within semesters or shorter periods. Even if they are implemented over longer time spans they are scaffolded with shorter modules. Changing personality traits does not fit well within this structure, whereas changing skills, which could more easily be organized into shorter time periods, would mesh better with typical educational structures. Asking teachers to teach how to perform SEB skills and students to demonstrate that they learned how to perform those skills is far more consistent with educational policy than asking teachers to change their students' personality traits.

## Limitations and future directions

The present study's strengths include its assessment of both SEB skills and personality traits, as well as its multimodal design that tested the ability of self-reported personality characteristics to predict an objectively scored measure of academic performance. However, the study also has several limitations that highlight promising directions for future research. First, we assessed only a subset of the BESSI skill facets in this study. Thus, our skill measures sample a narrower range of behavioral content than do their trait counterparts. In future research, the inclusion of the full BESSI (or an abbreviated version that systematically samples content from many facets) could provide a more comprehensive map of relationships between SEB skills and academic outcomes. Of course, a longer, more elaborated version of the Big Five should also be used if testing incremental validity remains the goal. Also, while the behavioral referents were very similar between the BESSI and BFI, they were not identical, which means that the constructs may have tapped slightly different content. It might be this slight difference in content that is responsible for the ability of the BESSI to provide incremental validity. Future research should test whether skill versus trait ratings of identical item content can be differentiated.

Second, we did not assess general cognitive ability as an additional predictor or control variable. It is therefore unclear whether SEB skills and personality traits differ in their degree of

overlap with intelligence, as well as their capacity to predict academic performance beyond cognitive ability. Third, we used the ACT composite score as an outcome variable. The high-stakes nature of this assessment may strengthen its relations with skills rather than traits, and comparative validity between SEB skills and personality traits could differ for different types of outcome variables in different contexts. For example, for predicting outcome variables that require continuous and enduring effort, such as grade point average, personality traits may outperform SEB skills. Moreover, it is unknown whether or not SEB skills outperform personality traits in non-academic settings, such as work settings. These lines of research questions are open for future research. Lastly, the sample for this study is not representative of high school students. Our sample consists of high achieving and mostly female students whose ACT scores were well above the national average for 2021 ACT scores. This sample characteristic could influence the results of this study. We believe that correlations between SEB skills and the achievement score are more likely to be higher for representative samples. For example, better faking ability for highly intelligent samples [22] and range restriction (i.e., restricted variability in participants) could both contribute to attenuation in effect sizes. This limitation warrants future research with more representative samples of students.

## Conclusion

The present study extends our understanding of SEB skills' and personality traits' incremental validity for predicting academic achievement. We demonstrated that BESSI skills are more highly associated with academic achievement compared to similar measures of personality traits. We look forward to future research testing the validity of SEB skills in predicting a diverse set of outcome variables in academic and industry settings.

## Supporting information

**S1 Checklist.**
(DOCX)

## Author Contributions

**Conceptualization:** Hee Jun Yoon, Brent W. Roberts.

**Data curation:** Christopher J. Soto.

**Formal analysis:** Hee Jun Yoon, Brent W. Roberts.

**Methodology:** Hee Jun Yoon, Brent W. Roberts, Madison N. Sewell, Christopher M. Napolitano, Christopher J. Soto.

**Project administration:** Brent W. Roberts.

**Resources:** Dana Murano, Alex Casillas.

**Software:** Hee Jun Yoon.

**Supervision:** Brent W. Roberts, Christopher M. Napolitano.

**Visualization:** Hee Jun Yoon.

**Writing – original draft:** Hee Jun Yoon, Brent W. Roberts, Madison N. Sewell, Christopher M. Napolitano, Christopher J. Soto.

**Writing – review & editing:** Brent W. Roberts, Madison N. Sewell, Christopher M. Napolitano, Christopher J. Soto, Dana Murano, Alex Casillas.

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
