## [Decision Letter · Decision Letter 0]

19 May 2023

PONE-D-23-09368Examining SEB skills’ incremental validity over personality traits in predicting academic achievementPLOS ONE

Dear Dr. Yoon,

Thank you for submitting your manuscript to PLOS ONE. After careful consideration, we feel that it has merit but does not fully meet PLOS ONE’s publication criteria as it currently stands. Therefore, we invite you to submit a revised version of the manuscript that addresses the points raised during the review process.

We look forward to receiving your revised manuscript.

Kind regards,

Hosam Al-Samarraie

Academic Editor

PLOS ONE

2. You indicated that ethical approval was not necessary for your study. We understand that the framework for ethical oversight requirements for studies of this type may differ depending on the setting and we would appreciate some further clarification regarding your research. Could you please provide further details on why your study is exempt from the need for approval and confirmation from your institutional review board or research ethics committee (e.g., in the form of a letter or email correspondence) that ethics review was not necessary for this study? Please include a copy of the correspondence as an ""Other"" file.

“No”

“No”

5. Please amend your manuscript to include your abstract after the title page.

6. Please include your tables as part of your main manuscript and remove the individual files. Please note that supplementary tables (should remain/ be uploaded) as separate "supporting information" files

Reviewers' comments:

Reviewer's Responses to Questions

**Comments to the Author**

1. Is the manuscript technically sound, and do the data support the conclusions?

Reviewer #1: Yes

Reviewer #2: Partly

2. Has the statistical analysis been performed appropriately and rigorously? 

Reviewer #1: Yes

Reviewer #2: No

3. Have the authors made all data underlying the findings in their manuscript fully available?

Reviewer #1: Yes

Reviewer #2: Yes

4. Is the manuscript presented in an intelligible fashion and written in standard English?

Reviewer #1: Yes

Reviewer #2: No

5. Review Comments to the Author

Reviewer #1: The present research, authors tested whether particular SEB skills predicted high school students’ ACT scores beyond their Big Five personality traits. Significant changes in the incremental regression coefficients were used as comparison parameters. The results enrich the predictors of academic achievement, and have certain reference significance for further guiding students' learning from the perspective of SEB, and test the validity of the SEB measurement questionnaire.

Therefore, I suggest accepting this manuscript.

Please revise the following aspects：

1.Supplement illustrate the theoretical and measured differences between SEB and Big Five personality traits and the need to predict academic achievement with SEB.

2.Soto, C. J., Napolitano, C., Sewell, M., Yoon, H. R., & Roberts, B. (2021). An Integrative Framework for Conceptualizing and Assessing Social, Emotional, and Behavioral Skills: The BESSI.

Soto, C. J. , Napolitano, C. M. , Sewell, M. N. , Yoon, H. J. , & Roberts, B. W. . (2022). An integrative framework for conceptualizing and assessing social, emotional, and behavioral skills: the bessi. Journal of personality and social psychology, 123(1), 192-222.

Reviewer #2: Review Comments

Title: Examining SEB skills’ incremental validity over personality traits in predicting academic achievement. Generally, the topic sounds interesting and could potentially contribute to the conceptualisation of the SEB skills. The objective(s) should be clearly stated. Providing a solid theoretical background through a review of the literature will help support the argument being put across. Research is clear on the relationship between traits, skills and performance. Therefore, it is still unclear why the authors are comparing traits to skills. In addition, consider including the potential contribution this study will make to theory and practice. Incorporating the following suggestions could help improve upon this manuscript.

Comment 1

On paragraph 2. Researchers ended abruptly on the distinction between behavioural traits and capabilities, which leaves readers hanging. It will be an interesting read if researchers could include how far the conversation on the potential difference between behavioural traits and capabilities has gotten.

Comment 2

Paragraph 5. Researchers should be able to clarify the conceptual difference between traits and skills, since their acquisition pathways differs.

Comment 3

The study is still not clear about the gap in knowledge it is seeking to fill and the possible contribution the outcome of the study will make to the theoretical development of the SEB concept.

Comment 4

Methods (participants and procedure)

The last sentence, …fort he… should read for the

Measures

Line 9: delete one of the “loading”

Demographic information: consider including the results of the control variables to justify why the significant findings are free from any undue influence.

Results (Correlations between the BESSI, BFI, and ACT)

Researcher should indicate the extent to which measures correlate with alternate measures of the same construct. You may consider indicating factor loadings and the AVE values for both convergent and construct validity and be guided by the threshold limits. Moreso, details on how discriminant validity of the measures were ascertained and the results would be of much help to this study.

Line 8: provide details of the results on the purported association between the Big Five traits and ACT instead of the general statement that several of the …

Incremental validity of SEB Skills and Big Five Traits

Since you are introducing a new concept “suppression artifacts”, it will be better to explain it either in the background or review of literature section.

Discussion

To improve the rigour of the study, researchers should include findings of existing research to support their results. Currently, only (Soto et al., in press) has been cited in the discussion. Discuss your findings in relation to existing knowledge on the subject matter.

References:

Researchers should ensure that referencing style conforms to the requirement of the Journal. In the meantime kindly check and ensure that Gerger, M… (2019), (ACT, 2019), (ACT, 2020) are properly cited and referenced.

6. PLOS authors have the option to publish the peer review history of their article (what does this mean?). If published, this will include your full peer review and any attached files.

Reviewer #1: No

Reviewer #2: No

---

## [Author Response · Author response to Decision Letter 0]

10 Nov 2023

Dear Dr. Al-Samarraie, 

We are submitting a revised version of our manuscript, entitled “Examining SEB skills’ incremental validity over personality traits in predicting academic achievement,” to be considered for publication in the PLOS ONE.

Thank you very much for inviting us to revise and resubmit this manuscript. Thanks also to you and the reviewers for your thoughtful and encouraging comments on the previous version. As detailed in the Revision Notes, we have incorporated all of your suggestions, and we sincerely believe that the revised manuscript is better for it. 

For financial disclosure, the authors received no specific funding for this work. The authors have declared that no competing interests exist.

Please feel free to let me know if you have any questions. 

We hope that you find the revised manuscript improved from the previous version and suitable for publication in PLOS ONE.

For the authors,

Hee Jun Yoon

Revision Notes

Points Raised by the Editor

Points Raised by Reviewer 1

R1.1. Supplement illustrate the theoretical and measured differences between SEB and Big Five personality traits and the need to predict academic achievement with SEB.

We agree that it is important to further explain the difference between SEB skills and personality traits, and therefore clarify the need for this research to examine whether SEB skills can add incremental validity over traits in predicting academic achievement. To incorporate this suggestion, we have added text to further explain the conceptual difference between skills and traits, and to clarify the value of predicting academic success from skills and traits (see pp. 3-7, as well as Points R2.1, R2.2, and R2.3, below). 

R1.2. Soto, C. J., Napolitano, C., Sewell, M., Yoon, H. R., & Roberts, B. (2021). An Integrative Framework for Conceptualizing and Assessing Social, Emotional, and Behavioral Skills: The BESSI.

Soto, C. J. , Napolitano, C. M. , Sewell, M. N. , Yoon, H. J. , & Roberts, B. W. . (2022). An integrative framework for conceptualizing and assessing social, emotional, and behavioral skills: the bessi. Journal of personality and social psychology, 123(1), 192-222.

As suggested, we have changed these citations with updated information.

Points Raised by Reviewer 2

R2.1. On paragraph 2. Researchers ended abruptly on the distinction between behavioural traits and capabilities, which leaves readers hanging. It will be an interesting read if researchers could include how far the conversation on the potential difference between behavioural traits and capabilities has gotten.

We agree that it is important to include the potential difference between behavioral traits and SEB skills. To incorporate this suggestion, we now further explain the conceptual distinction between personality traits and SEB skills in the introduction (see pp. 3-7).

R2.2. Paragraph 5. Researchers should be able to clarify the conceptual difference between traits and skills, since their acquisition pathways differs.

We agree that it is important to clarify this conceptual difference. As also suggested in Point R2.1 above, we have added a paragraph to clarify the conceptual difference between traits and skills in paragraph 5. 

R2.3. The study is still not clear about the gap in knowledge it is seeking to fill and the possible contribution the outcome of the study will make to the theoretical development of the SEB concept.

We agree that it is important to clarify the gap in knowledge we are seeking to fill and the possible contribution the outcome of the study will make to the theoretical development of the SEB skills. To incorporate this suggestion, we have added a paragraph on how SEB skills and personality traits may conceptually differ in predicting different educational outcomes, which further clarifies the possible contribution of this paper: 

“Moreover, conceptual differences between personality traits and SEB skills can manifest differently in predicting educational outcomes. Personality traits may best predict academic outcomes that reflect overall performance over time across assignments and exams, such as cumulative school grades, because personality traits represent consistent patterns of thought, feelings, and behaviors of individuals. On the other hand, SEB skills may play an especially important role in predicting standardized test scores because they represent capacities that people can purposefully employ when a specific, high-stakes situation calls for it. Therefore, even though traits and skills may both play an important role in predicting academic outcomes, they may play different roles for different outcomes.”

R2.4. Methods (participants and procedure). The last sentence, …fort he… should read for the

Thank you for catching this mistake. We have now fixed the clerical error. 

R2.5. Measures Line 9: delete one of the “loading”

Thank you. We have now deleted one of the “loading”.

R2.6. Demographic information: consider including the results of the control variables to justify why the significant findings are free from any undue influence.

We agree that it is important to include control variables to help justify why the significant findings are free from any undue influence. To incorporate this suggestion, Table 1 now provides precise p-value for the effects of the control variables, as well as the incremental validity of SEB skills and personality traits over and beyond the control variables in predicting ACT scores. To further clarify these relations, Table 2 now provides correlations between all variables. 

R2.7. Results (Correlations between the BESSI, BFI, and ACT)

Researcher should indicate the extent to which measures correlate with alternate measures of the same construct. You may consider indicating factor loadings and the AVE values for both convergent and construct validity and be guided by the threshold limits. Moreso, details on how discriminant validity of the measures were ascertained and the results would be of much help to this study.

We agree that establishing construct validity is important in psychological research. To incorporate this suggestion, the following part of the results section now reports correlations among measures: “Table 2 shows the means, standard deviations, and reliabilities as well as intercorrelations among all variables. Several notable results emerged. First, there was a general pattern of medium to strong intercorrelations among measures drawn from the same instrument. There were general positive correlations among the SEB skills, the Big Five traits, and the ACT composite score. Second, the convergent correlations between the SEB skills and their Big Five counterparts ranged from .48 to .XX and averaged .59, whereas the discriminant correlations averaged .27. This pattern of substantial—but far from perfect—associations supports SEB skills’ discriminant validity.” 

Moreover, in the introduction, we now cite a paper that directly examined the construct validity of SEB skills in relation to personality traits: Soto et al. (2022). 

R2.8. Line 8: provide details of the results on the purported association between the Big Five traits and ACT instead of the general statement that several of the …

We agree that it is important to report the associations between Big Five traits and ACT performance in detail. To incorporate this suggestion, Table 2 now reports the correlations of all five traits with ACT scores.

R2.9. Incremental validity of SEB Skills and Big Five Traits. Since you are introducing a new concept “suppression artifacts”, it will be better to explain it either in the background or review of literature section.

We agree that it is important to explain the concept of suppression artifacts. To incorporate this suggestion, we have added the following sentence: “In other words, these results indicate that Stress Regulation skill and Agreeableness trait are less positively associated with the ACT performance than would be expected from these constructs’ positive relations with Innovation skill.”

R2.10. Discussion. To improve the rigour of the study, researchers should include findings of existing research to support their results. Currently, only (Soto et al., in press) has been cited in the discussion. Discuss your findings in relation to existing knowledge on the subject matter.

We agree that it is important to synthesize the present findings with previous research. To incorporate this suggestion, we have added cited other recent studies indicating that skills provide unique information beyond traits: “This finding adds to a recent and growing body of evidence that skills can provide incremental validity over traits in predicting a broad range of consequential outcomes including academic engagement and achievement, occupational interests, social behaviors and relationships, and physical and psychological well-being (Breil et al., 2022, JOI; Soto et al., 2023, JRP; Soto et al., 2023, SPPS)” (see p. 17).

R2.11. References. Researchers should ensure that referencing style conforms to the requirement of the Journal. In the meantime kindly check and ensure that Gerger, M… (2019), (ACT, 2019), (ACT, 2020) are properly cited and referenced.

We agree that we should ensure the referencing style conforms to the requirement of PLOS ONE. We have therefore edited the manuscript to follow the format.

---

## [Editor Report · Decision Letter 1]

15 Dec 2023

Examining SEB skills’ incremental validity over personality traits in predicting academic achievement

PONE-D-23-09368R1

Dear Dr. Yoon,

We’re pleased to inform you that your manuscript has been judged scientifically suitable for publication and will be formally accepted for publication once it meets all outstanding technical requirements.

Kind regards,

Hosam Al-Samarraie

Academic Editor

PLOS ONE

Additional Editor Comments (optional):

Thanks for addressing all the concerns raised by the reviewers.
---

## [Editor Report · Acceptance letter]

22 Dec 2023

PONE-D-23-09368R1 

PLOS ONE

Dear Dr. Yoon, 

I'm pleased to inform you that your manuscript has been deemed suitable for publication in PLOS ONE. Congratulations! Your manuscript is now being handed over to our production team.

Kind regards, 

on behalf of

Dr Hosam Al-Samarraie 

Academic Editor

PLOS ONE